# Clinical Outcomes of Monolithic Zirconia Crowns with CAD/CAM Technology. A 1-Year Follow-Up Prospective Clinical Study of 65 Patients

**DOI:** 10.3390/ijerph15112523

**Published:** 2018-11-12

**Authors:** Ioannis Konstantinidis, Dimitra Trikka, Spyridon Gasparatos, Miltiadis E. Mitsias

**Affiliations:** 1K Clinic. Marathonos 60 Avenue, Anixi, GR14569 Athens, Greece; konstanioannis@hotmail.com; 2Dental Center, 29 Marasli Street, Kolonaki, GR10676 Athens, Greece; mirka.trikka@gmail.com (D.T.); gasparatosdent@gmail.com (S.G.); 3Department of Periodontology & Implant Dentistry, New University College of Dentistry, New York, NY 10010, USA

**Keywords:** zirconia, crowns, USPHS criteria, CAD/CAM, monolithic restorations, success rate, fracture, prospective study

## Abstract

**Aim/Purpose:** The primary aim of this study was to examine the clinical performance of posterior monolithic single crowns in terms of failure or complications and the secondary aim was to assess the quality of these restorations according to the United States Public Health Service (USPHS) criteria. **Methods:** Ιn a private dental clinic, 65 patients with need of posterior crowns were restored with monolithic zirconia crowns. All the restorations were evaluated 6 and 12 months after their cementation. The modified United States Public Health Service (USPHS) criteria and periodontal parameters were applied for the clinical evaluation of the crowns. Restorations with Alpha or Bravo rating were considered a success. **Results:** Descriptive statistics and nonparametric tests were used for statistical analysis. Sixty-five patients (mean age: 49.52) were restored with 65 monolithic zirconia crowns. No fracture of the restorations was recorded. The overall success rate was 98.5%. The clinical quality of all crowns was acceptable except for the marginal discoloration of one crown at the 6- and 12-month follow-up examination. **Conclusions:** In this study, no fracture of single-tooth monolithic crowns occurred and the success rate was high. Monolithic zirconia restorations fabricated is a viable option for the restoration of single posterior teeth.

## 1. Introduction

The CAD/CAM technology allows the fabrication of esthetic restorations with high efficiency in daily practice [1]. With the application of the digital workflow in daily practice, it is possible to achieve high precision even with high-strength materials like lithium disilicate and zirconia. [2] Zirconia-based single crowns (SCs) and fixed dental prostheses (FDPs) have earned much popularity due to their mechanical properties [3,4], excellent biocompatibility [5], and aesthetics. Studies have shown that the survival rates of zirconia-based single crowns are high and they are comparable to metal-ceramic FDPs [6]. However, the success of metal- or zirconia-ceramic restorations is diminished through the frequent occurrence of chippings [7,8]. This technical complication affects the aesthetic outcome, function, and longevity of the restoration.

In order to overweigh this limitation, the use of monolithic zirconia FDPs was proposed. The use of monolithic zirconia FDPs for the restoration of missing teeth requires only minimal tooth reduction since there is no need of space clearance for the veneering material. In addition, the clinician takes advantage of the precision of the CAD/CAM technology, which allows the milling of the SCs in full contour. An advantage of these restorations is also that the cost and time for the production is significantly lower in comparison to porcelain-fused-to-zirconia SCs. The monolithic approach for the restoration of single teeth is not new. This approach was introduced in order to avoid the technical problems that were associated with veneering. Prospective clinical trials showed high survival rates [9,10]. Monolithic zirconia crowns have the significant advantage in comparison to the monolithic lithium disilicate crowns, that the required tooth substance removal is significantly less for the zirconia crowns, due to the high fracture resistance that zirconia has [11].

Using high-strength materials that can be characterized with glazes and colors to mimic natural esthetics, monolithic zirconia restorations seem to be a promising alternative to metal- and zirconia-veneered restorations.

However, there is limited evidence available concerning the clinical behavior of monolithic zirconia restorations [12]. Laboratory studies have shown the high mechanical properties of these restorations. Sun et al. and Johansson et al. have demonstrated that the monolithic zirconia crowns have superior fracture resistance when compared to monolithic lithium disilicate crowns, layered zirconia crowns and metal ceramic crowns [13,14]. These results are in accordance with the recent results of an in vitro study presented by Nakamura et al., who reported that the fracture load of monolithic zirconia crowns with 0.5 mm of thickness was higher than that of monolithic lithium disilicate crowns with 1.5 mm of thickness, respectively [15].

Although the use of an extremely hard material offers advantages for the fracture resistance of the restoration, the clinician may be concerned about the wear of the opposing dentition in the long term. Stober et al. showed in a clinical study that the enamel wear of teeth opposing to monolithic zirconia crowns after 6 months was significantly higher than the contralateral natural antagonists [16].

The primary aim of this study was to examine the clinical performance of posterior monolithic single crowns in terms of failure or complications and the secondary aim was to assess the quality of these restorations according to the United States Public Health Service (USPHS) criteria.

## 2. Materials and Methods

The study was carried out in accordance with the ethical principles of the World Medical Association Declaration of Helsinki. All patients were members of a private dental center in Athens, Greece (www.dental-center.eu). The inclusion criteria in order for a patient to participate in the study were: a signed consent form, a first or a second molar, a first or a second premolar which required a monolithic restoration either with/or without an endodontic therapy.

The exclusion criteria were a severe periodontal status of the patient, the appearance of a parafunction habit-like bruxism [17], age limitation of less than 18 years old, and a condition like a pregnancy and/or lactation.

To each participant was delivered one crown only. All patients were restored with Zirconia Monolithic restorations. All restorations were performed by the same dentist (M.M.) and the same laboratory and technician (M.P.).

Tooth preparations and the respective schemes followed all the requirements that a monolithic ceramic crown necessitates. Circumferential axial and occlusal reduction of the prepared teeth were modified to the number of the tooth that needed the crown. Monolithic crowns require a minimum wall thickness of 1 mm. For the impression procedure, the gingivae were displaced with retraction cords (Ultrapak; Ultradent, South Jordan, UT, USA). A two-cord placement was followed. The two displacement cords were packed as following: First the 000 and then the 00 were used to displace the gingivae accordingly (Figure 1). Impression was taken using a Vinyl-Poly Siloxane material (Aquasil Ultra LV; Dentsply, York, PA, USA). Following this, the later stone models were scanned by a scanner (3Shape D700, Holmens Kanal 7, Copenhagen, Denmark). Design of the crowns was produced using the 3Shape dental design software. Manufacturing was done first by Zenotec CAM by Wieland Dental and then milling followed by CNC machine Wieland Select (Wieland Dental, Pforzheim, Germany). The block that was used was the Zenostar translucent blank (Wieland, Oakville, ON, Canada). The sintering process was completed by Wieland cube furnace. Staining and glazing was finished by IPS e.max stain. (Crystall/Glaze; Ivoclar Vivadent AG, Schaan, Liechtenstein; and Programat CS; Ivoclar Vivadent AG) before placement (Figure 2a,b)

The process that the team followed was not a one-day procedure but a classic procedure with the use of provisional acrylic crowns (Structur 2 SC; Voco GmbH, Cuxhaven, Germany) until the final ones were delivered to the patients after 7 days from the impression phase. After removal of the provisional crowns, permanent crowns were seated and, as needed, they were fitted by initially confirming the interproximal contacts and then validating the internal fit with additional silicone (black for ceramic, white for MC; Fit-Checker; GC America Inc., Alsip, IL, USA) to disclose and verify the fit of the intaglio surface. Concerning the cementation of the crowns, the following steps were prepared: Restorations surfaces were sandblasted with 50 micron aluminum oxide at a pressure of 2 Bar (30 psi) to create a matte surface appearance. After the crowns were cleaned with alcohol and air dried with oil-free air, they were silanized for 1 min (Monobond Plus, Ivoclar Vivadent AG). The dental–teeth surfaces were initially etched using 37% phosphoric acid (Total Etch; Ivoclar Vivadent AG) for 30 s and then treated for another 30 s with adhesive bonds Multilink Primer A + B. Then the crowns were inserted and luted using the Multilink dual cure resin cement (Multilink; Ivoclar Vivadent AG). In accordance with the manufacturer’s recommendations. Remnants and excess luting material was removed. Last occlusal adjustments were evaluated and adjusted accordingly, if needed.

### 2.1. Evaluation Procedure

For the evaluation of the single crowns at the baseline and the recall appointments, the US Public Health Service (USPHS) criteria were used. (Table 1) Judgment occurred by one Prosthodontist only (M.M.). More specifically color, marginal fit, marginal discoloration, secondary caries, surface texture, gross fracture were recorded and evaluated. According to how successful the latter were, they were categorized into Alpha, Bravo, or Charlie. All parameters were rated Alpha (A) in case of no problem, Bravo (B) in case of minor extent of the complication, and Charlie (C) if the complication was major or if the restoration had to be removed due to the complication. The periodontal evaluation was assessed by determining the plaque scone (PI) and bleeding on probing (BOP) at the abutment and the control teeth.

### 2.2. Study Outcomes

The primary outcome of this study was to examine the survival rate of the monolithic crowns during the observation period. Any fracture of the restorations was considered as failure. The secondary outcome was to evaluate the success of the restorations according to the USPHS criteria. Restorations with “Alpha” and “Bravo” rates were considered successful.

### 2.3. Statistical Analysis

Descriptive analysis was performed for the evaluation of the restoration and the tooth outcome, according to the modified USPHS criteria. The Wilcoxon singed-rated test was used to compare differences in periodontal parameters between test and control teeth.

## 3. Results

Sixty-five patients with a mean age of 49.52 years were restored with 65 monolithic zirconia crowns. Respectively, 29 (44.6%) crowns were placed in the maxilla and 36 on the mandible. From those, 46 crowns were root canal treated teeth. All patients attended the 6-month and the 12-month examination. During the observation period, all the crowns were intact, resulting in a survival rate of 100% (Figure 3).

Regarding the quality assessment of the restorations, with the exception of one restoration which rated with Charlie for marginal discoloration at the 6- and 12-month examination, all the other restorations were rated either with Alpha or Bravo. This resulted in a success rate of 98.5%.

At the baseline and the 6-month evaluation, all crowns obtained an Alpha for the criteria: “secondary caries” and “gross fracture” Figure 4). The percentages of restorations with B rating for marginal discoloration increased from 4.6% at the baseline to 16.88 at the 6- and 12-month examination.

Regarding the “surface texture”, the B rating changed from 1.5% to 6.2% at the 6-month examination, and then remained at the same percentage at the 12-month examination. From the 6-month to the 12-month examination, most of the percentages of A and B rates did not change. A slight increase of B rates of anatomic contour between the 6- (3.1%) and the 12-month (7.7%) examination was detected (Table 2).

One of the parameters used to evaluate the periodontal health was the BOP. Although there was a significant decrease of the BOP full-mouth score from the baseline to the 12-month examination, the BOP score at the test teeth was increased significantly from 0.0 at the baseline examination to 1.8 at the 12-month examination. It was also found that the BOP score at the control decreased significantly from 7.79 at the baseline to 3.34 at the 12-month examination (Table 3).

Regarding the plaque score, there was no difference within the groups during the whole observation period. However, the plaque score at the abutment teeth was significantly lower at the baseline and 6-month examination (Table 4). At the 12-month examination, there was no difference in the plaque score between abutment and control teeth.

## 4. Discussion

In this prospective study of monolithic zirconia crowns restoring single posterior teeth, no crown fractures, no loss of retention, and no tooth loss were found, resulting in a high survival rate. Furthermore, the technical outcomes evaluated with the USPHS criteria were very satisfactory as well. According to the USPHS criteria, the clinical quality of all crowns were in an acceptable and pleasing range except for the marginal discoloration of one crown. In this case, the required adjustments were performed and there was no need for replacement of the crown. Monolithic zirconia crowns have high fracture resistance and this allows the tooth restoration without excessive tooth preparation. This is one of the reasons monolithic zirconia crowns have become a reliable treatment alternative to porcelain-fused-to-metal and veneered crowns. The high survival rates shown in our study are also supported from the results of other studies [18,19,20].

In this study, no secondary caries could be detected. The development of caries could not be found in similar studies. The good marginal fit of the monolithic zirconia crowns prevents the development of caries. However, in vitro studies have shown that the marginal adaptation of the monolithic crowns can be affected by several factors. Particularly, Hamza et al. showed that different CAD/CAM systems show different marginal discrepancy of the monolithic crowns and Kale et al. showed that the marginal fit of the monolithic crowns can be affected by the cementation process, but it was within the acceptable range (<120 μm) [21,22].

Batson et al. compared the marginal adaptation between metal ceramic, lithium disilicate, and monolithic single crowns and they presented that the monolithic crowns have significantly better marginal integration in comparison to lithium disilicate crowns [18].

Considering the technical outcomes rated by USPHS criteria, no significant differences were found between the follow-up examinations. Of the restorations, 15% were rated with Bravo for color match. The missing translucency and the bright opacity prevent the successful color match. The slight mismatch of color was already detected at the time of final crown delivery. Despite this color mismatch, the patients were satisfied with the color and decided for the final insertion. Similar findings were also reported from Worni et al., and this result highlights the difficulty to achieve the desired outcome only by superficial shading [19].

In our study, the surface texture of the restorations rated with Bravo increased by 4.7%. There are several factors that can affect the surface texture like tooth brushing, abrasion, or attrition [23,24]. With the introduction of monolithic restorations in daily practice, there was a lot of concern about the damage of the hard zirconia surface on the antagonist enamel. Results of RCTs and in vitro studies have shown that the wear of opposing enamel is less with monolithic zirconia as compared to the wear of other crowns [25,26].

Regarding the periodontal parameters, the results of our study showed a significant increase of BOP score in the teeth supporting the single crowns. This is in contrast to the results of studies of Worni et al. and Batson et al. that also showed increased BOP score and increased CGF on teeth supporting the monolithic zirconia crowns than on nonrestored teeth; however, the difference was not statistically significant [18,19]. On the other hand, the plaque score did not differ between restored and nonrestored teeth. A possible explanation for this finding could be that, in the majority of the crowns in our study, the margins of the restorations were subgingivally. The indication for the subgingival placement of the margins was the presence of old fillings or caries at the survival area of the restored teeth. Clinical studies of Paniz et al. 2016 and Gemalmaz et al. 2002 have also shown that the subgingival margins of the crowns are associated with increased bleeding on probing [27,28].

The present study has some limitations (like the number of patients included, the limited number of crowns examined and the short follow-up period). Another limitation is that the workflow was not completely digital since the impressions were not taken with intraoral scanners. Studies have shown that the intraoral scanners demonstrate high accuracy both with natural teeth and implants [29,30].

## 5. Conclusions

The results of the present study should be examined in a critical light because of the small study population and the small evaluation period. Despite the limitation of this study, it can be concluded that the monolithic zirconia restorations fabricated with CAD/CAM technology is a viable option for the restoration of single posterior teeth. This study revealed no fracture of single-tooth monolithic crowns and the success rate was high. However, long-term randomized control studies with a large sample of patients are required to adequately document possible benefits of monolithic zirconia restorations in comparison to other treatments.

## Figures and Tables

**Figure 1 ijerph-15-02523-f001:**
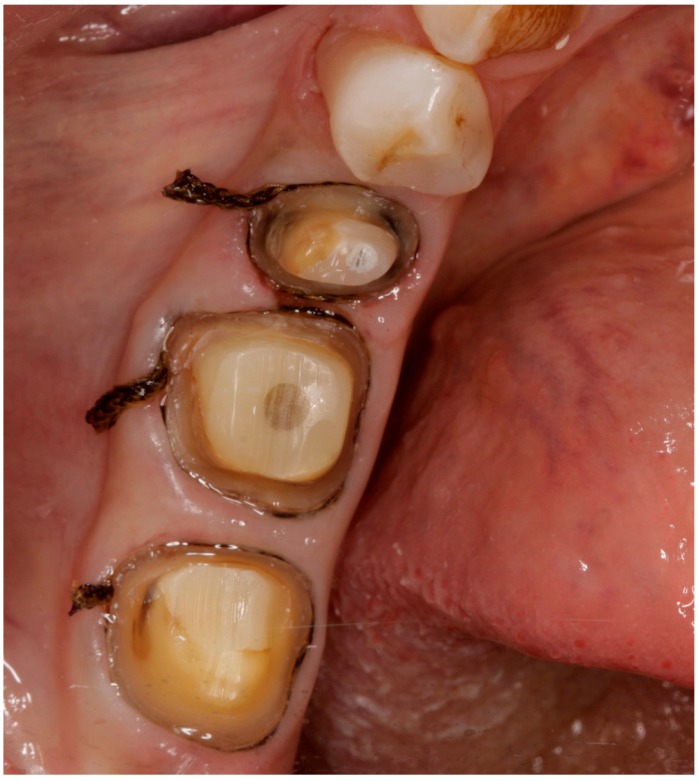
Impression technique for monolithic zirconia restorations. Only the second premolar was included in our study.

**Figure 2 ijerph-15-02523-f002:**
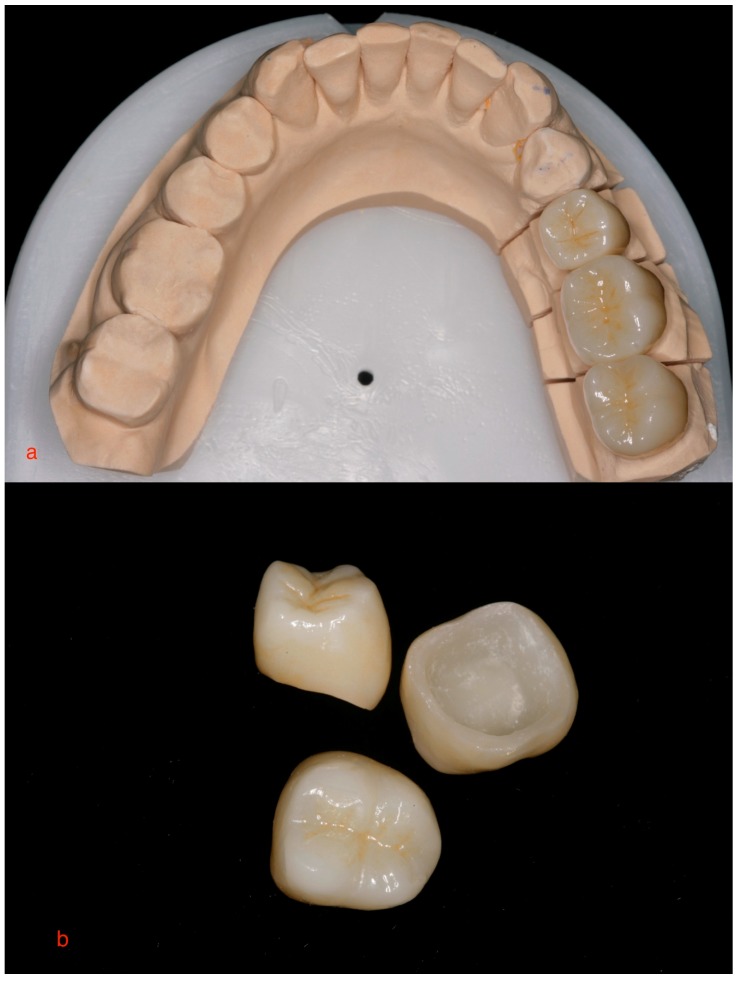
(**a**,**b**). Final monolithic zirconia crowns with CAD/CAM technology.

**Figure 3 ijerph-15-02523-f003:**
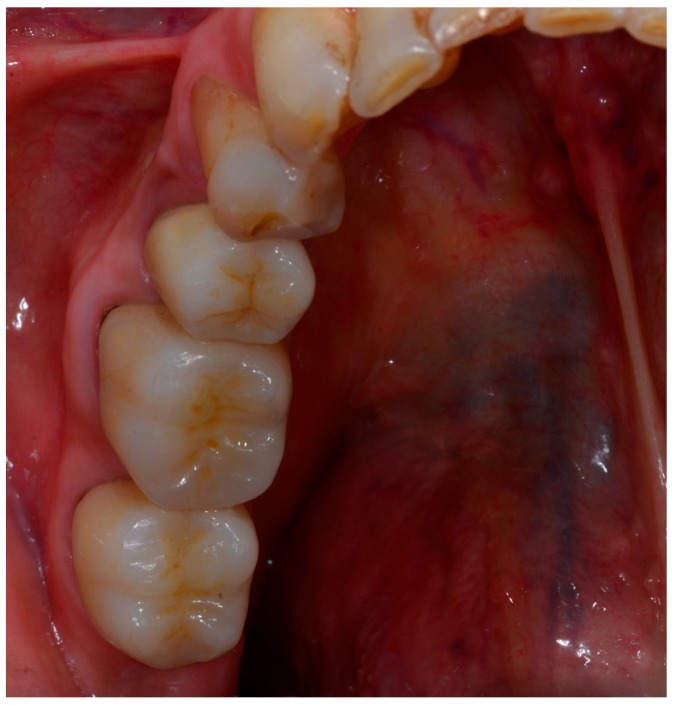
Monolithic zirconia crowns (1-year follow-up).

**Figure 4 ijerph-15-02523-f004:**
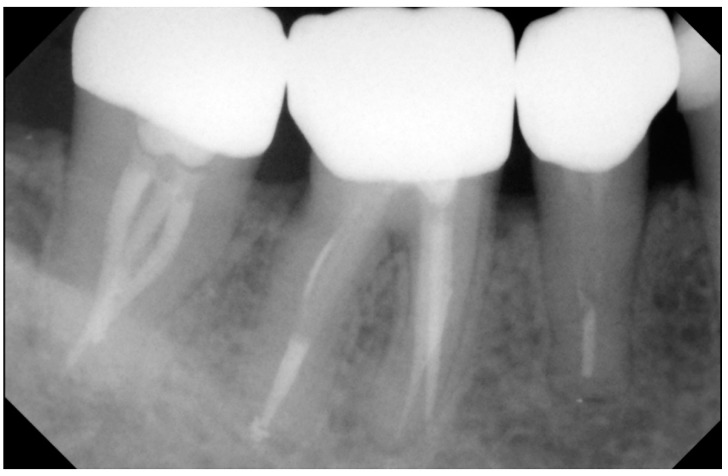
Periapical *x*-ray (1-year follow-up).

**Table 1 ijerph-15-02523-t001:** U.S. Public Health Service criteria.

Parameters	Alpha (A)	Bravo (B)	Charlie (C)
Color Match	The restoration appears to match the shade and the translucency of adjacent tooth tissues. (visual inspection)	The restoration does not match the shade and the translucency of adjacent tooth tissues, but the mismatch is within the normal range of the tooth shades. (within normal range: Similar to silicate cement restorations for which dentist did not quite succeed in the matching tooth color by his choice among available silicate cement shades). (visual inspection)	The restoration does not match the shade and the translucency of the adjacent tooth structure, and the mismatch is outside the normal range of the tooth shades and translucency. (visual inspection)
Anatomic contour	The restoration is a continuation of existing anatomic form or is slightly flattened. It may be overcontoured. When the side of the explorer is placed tangentially across the restoration, it does not touch two opposing cavosurface line angles at the same time. (visual inspection and explorer)	A surface concavity is evident. When the side of the explorer is placed tangentially across the restoration, it does not touch two opposing cavosurface line angles at the time, but the dentin or base is not exposed. (visual inspection and explorer)	There is a loss of restorative substance such that a surface concanity is evident and the base and/or dentin is exposed. (visual inspection and explorer)
Cavosurface marginal discoloration	There is no visual evidence of marginal discoloration different from the color of the restorative material and from the color of the adjacement. (visual inspection)	There is visual evidence of marginal discoloration at the junction of the tooth structure and the restoration, but the discoloration has not penetrated along the restoration in a pulpal direction. (visual inspection)	There is visual evidence of marginal discoloration at the junction of the tooth structure and the restoration that has penetrated along the restoration in a pulpal direction. (visual inspection)
Marginal Integrity	The explorer does not catch when drawn across the surface of the restoration toward the tooth, or, if the explorer does not catch, there is no visible crevice along the periphery of the restoration. (visual inspection and explorer)	The explorer catches and there is visible evidence of the crevice, which the explorer penetrates, indicating that the edge of the restoration does not adapt closely to the tooth structure. The dentin and/or the base is not exposed, and the restoration is not mobile. (visual inspection and explorer)	The explorer penetrates crevice defect extended to the dento-enamel junction. (explorer)
Secondary caries	The restoration is a continuation of existing anatomic form adjacent to the restoration. (visual inspection)	There is visual evidence of the dark keep discoloration adjacent to the restoration (but not directly associated with cavosurface margins). (visual inspection and explorer)	
Surface texture	Surface texture similar to polished enamel as determined by means of a sharp explorer. (explorer)	Surface texture gritty or similar to a surface subjects to a white stone or similar to a composite containing supramicron-sized particles. (explorer)	Surface pitting is sufficiently coarse to inhibit the continuous movement of an explorer across the surface. (explorer)
Gross fracture	Restoration is intact and fully retained.	Restoration is partially retained with some portion of the restoration still intact.	Restoration is completely missing.

**Table 2 ijerph-15-02523-t002:** Quality assessment of the restoration at baseline, 6 months, and 12 months follow-up.

	Baseline			6 Months			12 Months		
	A	B	C	A	B	C	A	B	C
Color match	87.7%	12.3%	0	84.6%	15.4%	0.0%	84.6%	15.4%	0.0%
Marginal discoloration	95.4%	4.6%	0	83.1%	16.88%	0.02%	83.1%	16.88%	0.02%
Secondary caries	100%	0.0%	0	100%	0.0%	0.0%	100%	0.0%	0.0%
Anatomic form	96.9%	3.1%	0	96.9%	3.1%	0.0%	92.3%	7.7%	0.0%
Marginal integrity	93.8%	6.2%	0	92.3%	7.7%	0.0%	93.8%	6.2%	0.0%
Surface texture	98.5%	1.5%	0	93.8%	6.2%	0.0%	96.9%	3.1%	0.0%
Gross fracture	100%	0.0%	0	100%	0.0%	0.0%	100%	0.0%	0.0%

**Table 3 ijerph-15-02523-t003:** BOP at baseline, 6 months, and 12 months follow-up.

BOP	Baseline	6 Months	12 Months
Abutment tooth	0.0	3.62	1.8
Control tooth	7.96	6.43	3.34
Full-mouth	6.20	5.35	3.75

**Table 4 ijerph-15-02523-t004:** PI at baseline, 6 months, and 12 months follow-up.

PI	Baseline	6 Months	12 Months
Abutment tooth	3.12	9.43	6.11
Control tooth	19.48	15.92	10.55
Full-mouth	19.48	15.58	13.80

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
