# Peer review of "Clinical Outcomes of Monolithic Zirconia Crowns with CAD/CAM Technology. A 1-Year Follow-Up Prospective Clinical Study of 65 Patients"

_ijerph, 2018, doi:10.3390/ijerph15112523_

Round 1
Reviewer 1 Report
in the title it is not clear how many patients you have treated and the follow-up. this manuscript should have an abstract structured in sections to facilitate the readers, the authors should not forget to define their variables in the methods and should report their results in order of importance accordingly. in the methods section the variables section is missing and the Declaration of Helsinki on Experimentation on human subjects is missing (this is important in human studies). in the results no figures and figures should be added, you should moreover clarify your poor biologic results that are in contrast with the evidence reported by other studies in which zirconia performed very well. this work pathway is not a full digital workflow because you did not use intraoral scanners. tables are correct. english form should be checked because there are some sentences that do not read well, use a proofreading site to do this.
Author Response
Dear Reviewer,
Thank you for your very careful review of our paper, and for the comments, corrections and suggestions that ensued. A major revision of the paper has been carried out to take all of them into account.
Point 1: in the title it is not clear how many patients you have treated and the follow-up.
The title is now changed to: “Clinical outcomes of monolithic zirconia crowns with CAD/CAM technology. A 1-year follow-up prospective clinical study of 65 patients.”
Point 2: this manuscript should have an abstract structured in sections to facilitate the readers,
Your comment is now addressed. The abstract is now structured into aim/purpose, methods, results and conclusions.
Point 3: the authors should not forget to define their variables in the methods
The variables are now detailed described in the methods section.
“More specifically color, marginal fit, marginal discoloration, secondary caries, surface texture, gross fracture were recorded and evaluated. According to how successful the later were they categorized in to Alpha, Bravo or Charlie. All parameters were rated Alpha (A) in case of no problem, Bravo (B) in case of minor extent of the complication and Charlie (C), if the complication was major or if the restoration had to be removed due to the complication. The periodontal evaluation was assessed by determing the plaque scone (PI) and bleeding on probing (BOP) at the abutment and the control teeth.
Study Outcomes
The primary outcome of this study was to examine the survival rate of the monolithic crowns during the observation period. Any fracture of the restorations was considered as failure. The secondary outcome was to evaluate the success of the restorations according to the USPHS criteria. Restorations with “Aplha” and “Bravo” rates were considered successful.”
Point 3: “should report their results in order of importance accordingly”.
Now the results are presented in the following order:
Survival> Success> Outcomes according to USPHS criteria> Periodontal outcomes
Point 4: in the methods section the variables section is missing
The variables are now described in the methods section (see also Point 2).
Point 5: the Declaration of Helsinki on Experimentation on human subjects is missing (this is important in human studies).
We have now stated that this study was conducted according to the Declaration of Helsinki for Medical studies. The Declaration of Helsinki includes ethical approval as a component.
Point 6: in the results no figures and figures should be added,
We have now added one case.
Point 7: you should moreover clarify your poor biologic results that are in contrast with the evidence reported by other studies in which zirconia performed very well.
We agree with your comment. We were quiet surprised from these findings. The issue in now addressed:
A possible explanation for this finding could be that in the majority of the crowns in our study the margins of the restorations were subgingivally. The indication for the subgingival placement of the margins was the presence of old fillings or caries at the cervival area of the restored teeth. Clinical studies of Paniz et al. 2016 and Gemalmaz et al. 2002 have also shown that the subgingival margins of the crowns are associated with increased bleeding on probing. [27,28].
Point 8: this work pathway is not a full digital workflow because you did not use intraoral scanners.
This is correct and we agree that this could be a limitation of the study. The following part is added in the results section:
The present study has some limitations (like the number of patients included, the limited number of crowns examined and the short follow-up period). Another limitation is that the workflow was not completely digital since the impressions were not taken with intraoral scanners. Studies have shown that the intraoral scanners demonstrate high accuracy both with natural teeth and implants [29,30].
Point 9: english form should be checked because there are some sentences that do not read well, use a proofreading site to do this.
Writing mistakes are now corrected.
Reviewer 2 Report
title
- please erase the term "fabricated" from the title, as well as from the abstract
abstract
- in the abstract please state both your primary and secundary aims or purposes (copy the end of intro).
the secundary needs to be better explained in the methods because people (readers) are not aware of these indexes
- the abstract should be structured into aim/purpose, methods, results, conclusions, please correct it accordingly
- line 21 abstract, "endodontic complications" have nothing to do with monolithic zirconia crowns, please erase
- it is not clear in the abstract what do you mean for "success". please define. survival is clear, success no.
- line 23, "the biological integration of the zirconia crowns was not favourable". this sentence is against the current
evidence of almost all scientific literature, i feel it is mandatory to reformulate here or erase it. if you write this,
all your conclusions are completely nonsense
intro
- very good intro specially the first three sentences
- line 6, please erase the term "Meta-analysis" and insert there "Studies"
- line 41, Sailer et al, please erase this sentence because it should not be here
- i do not understand why, if the present paper deals with single crowns, in your intro after line 42 you start to talk about FDPs.
please erase and focus on single crowns. from line 50 all is acceptable, you should just re-organize your intro structure in
order to connect all these parts
methods
- well written, and clear, but here you should better and more extensively define your study outcomes, in a dedicated chapter
before the chapter of statistical analysis. please introduce a chapter named "study outcomes" where you define them.
put the things in logical order
results
- results should be re-organized, in order to follow the order established in the methods- outcome variables.
they should report in logical order what you obtained, with regard to the outcome variables. please rewrite
- no clinical pictures? i expect you to add at least 1 complete clinical case, including the cad screenshots and most
of all, the clinical and - if possible - radiographic controls. one case is mandatory, high quality pictures are needed
discussion
-line 167 the misleading sentence regarding the biological integration. zirconia is an excellent material as reported
in many studies and literature reviews. please reformulate or erase this sentence. please reformulate from line 196 as well.
these sentences are not acceptable. which kind of zirconia did you use here? Wieland? please provide this info
in the methods.
- no sentences at the end of your discussion regarding the limits of this clinical studies that are evident. please add
here this sentence (limited n° patients, limited n° crowns, no follow-up, no direct digitization, only clinical control, etc)
direct digitization with intraoral scanners is probably the best option now because of high accuracy of the scanners
itselves- you should state clearly this limit of your study, with intraoral scanners the workflow could be probably
better - please add more references here there are several studies that demostrates accuracy of intraoral scanners
both with natural teeth PMID: 29471825 and with implants PMID: 30112398 please add these references
conclusions
- state here the limits of your study and report that further studies are needed with a longer follow-up time,
more patients etc
references
- you should add at least two more references at the end of your discussion when you state the limits of indirect digitization
PMID: 29471825 - PMID: 30112398 add these ones
tables
- ok
figures
- figures are missing please add at least one beautiful case
Author Response
Dear Reviewer 2,
Thank you for reviewing our manuscript and for providing supportive comments. We appreciate the effort you made to improve this manuscript and are grateful for your insightful comments.
Point 1: please erase the term "fabricated" from the title, as well as from the abstract
Now, the title of the paper does not include the term “fabricated”.
Clinical outcomes of monolithic zirconia crowns with CAD/CAM technology. A 1-year follow-up prospective clinical study of 65 patients.
Point 2: in the abstract please state both your primary and secondary aims or purposes (copy the end of intro).
The primary and secondary outcomes were added in the abstract section.
Aim/purpose: The primary aim of this study was to examine the clinical performance of posterior monolithic single crowns in terms of failure or complications and the secondary aim was to assess the quality of these restorations according to the United States Public Health Service (USPHS) criteria.
Point 3: the secondary needs to be better explained in the methods because people (readers) are not aware of these indexes
We described the criteria for all the outcomes and we included a Table that explains the rating of the USPHS criteria (Table A).
Point 4: the abstract should be structured into aim/purpose, methods, results, conclusions, please correct it accordingly
The abstract is now structured accordingly.
Point 5: line 21 abstract, "endodontic complications" have nothing to do with monolithic zirconia crowns, please erase
Thank you for the careful observation. The term “endodontic complications” has been removed.
Point 6: it is not clear in the abstract what do you mean for "success". please define. survival is clear, success no.
We defined the criteria for success and survival in the abstract and methods section.
The primary outcome of this study was to examine the survival rate of the monolithic crowns during the observation period. Any fracture of the restorations was considered as failure. The secondary outcome was to evaluate the success of the restorations according to the USPHS criteria. Restorations with “Aplha” and “Bravo” rates were considered successful.
Point 7: "the biological integration of the zirconia crowns was not favourable". this sentence is against the current evidence of almost all scientific literature, i feel it is mandatory to reformulate here or erase it. if you write this, all your conclusions are completely nonsense
We definitely agree with your statement. Many studies have shown the good biological integration of zirconia in comparison to other materials. We have to admit, that we were also surprised by the results. However, we tried to explain these results. Having these findings do not necessarily means that zirconia is an inadequate material, but we believe the negative findings resulted for co-factors. As we explain in the manuscript the subgingival placement of the crown margin could be a reason for these findings. This has been also reported in other clinical studies. ( PMID: 11854676, PMID: 26445857)
If you agree with that, we would like to include this outcome as part of our results. If we expected only positive periodontal outcomes, we wouldn’t have included the periodontal parameters in the first place. We don’t believe that not reporting the clinical outcomes will increase the scientific value of the paper.
Point 8: please erase the term "Meta-analysis" and insert there "Studies"
We changed the term accordingly.
Point 9: line 41, Sailer et al, please erase this sentence because it should not be here
We removed the whole sentence according to your comment.
Point 10: i do not understand why, if the present paper deals with single crowns, in your intro after line 42 you start to talk about FDPs. please erase and focus on single crowns. from line 50 all is acceptable, you should just re-organize your intro structure in order to connect all these parts
We changed the introduction according to your suggestions.
Point 11: well written, and clear, but here you should better and more extensively define your study outcomes, in a dedicated chapter before the chapter of statistical analysis. please introduce a chapter named "study outcomes" where you define them. put the things in logical order.
We added the following part:
For the evaluation of the single crowns at the baseline and the recall appointments the US Public Health Service (USPHS) criteria were used. (table A) Judgment occurred by one Prosthodontist only (M.M). More specifically color, marginal fit, marginal discoloration, secondary caries, surface texture, gross fracture were recorded and evaluated. According to how successful the later were they categorized in to Alpha, Bravo or Charlie. All parameters were rated Alpha (A) in case of no problem, Bravo (B) in case of minoer extent of the complication and Charlie (C), if the complication was major or if the restoration had to be removed due to the complication. The periodontal evaluation was assessed by determing the plaque scone (PI) and bleeding on probing (BOP) at the abutment and the control teeth.
Study Outcomes
The primary outcome of this study was to examine the survival rate of the monolithic crowns during the observation period. Any fracture of the restorations was considered as failure. The secondary outcome was to evaluate the success of the restorations according to the USPHS criteria. Restorations with “Aplha” and “Bravo” rates were considered successful.
Point 12: results should be re-organized, in order to follow the order established in the methods- outcome variables. they should report in logical order what you obtained, with regard to the outcome variables. please rewrite
Now the results are presented in the following order:
Survival> Success> Outcomes according to USPHS criteria> Periodontal outcomes
Point 13: no clinical pictures? i expect you to add at least 1 complete clinical case, including the cad screenshots and most of all, the clinical and - if possible - radiographic controls. one case is mandatory, high quality pictures are needed
We have now added one case.
Point 14: line 167 the misleading sentence regarding the biological integration. zirconia is an excellent material as reported in many studies and literature reviews. please reformulate or erase this sentence. please reformulate from line 196 as well. Which kind of zirconia did you use here? Wieland? please provide this info in the methods.
Please consider Point 7.
This is in contrast to the results of studies of Worni et al. and Batson et al. that showed also increased BOP score and increased CGF on teeth supporting the monolithic zirconia crowns than on non-restored teeth; however, the difference was not statistically significant[19,18]. A possible explanation for this finding could be that in the majority of our study the margins of the restorations were subgingivally. On the other hand, the plaque score did not differ between restored und non-restored teeth.
We used Wieland. This information is included in the methods section.
Manufacturing was done first by Zenotec CAM by Wieland Dental and then milling followed by CNC machine Wieland Select (Wieland Dental 75175 Pforzheim, Germany). The block that was used was the Zenostar translucent blank (Wieland).
Point 15: no sentences at the end of your discussion regarding the limits of this clinical studies that are evident. please adhere this sentence (limited n° patients, limited n° crowns, no follow-up, no direct digitization, only clinical control, etc) direct digitization with intraoral scanners is probably the best option now because of high accuracy of the scanners itselves- you should state clearly this limit of your study, with intraoral scanners the workflow could be probably better - please add more references here there are several studies that demonstrates accuracy of intraoral scanners both with natural teeth PMID: 29471825 and with implants PMID: 30112398 please add these references
The following part was added according to your suggestions
The present study has some limitations (like the number of patients included, the limited number of crowns examined and the short follow-up period). Another limitation is that the workflow was not completely digital since the impressions were not taken with intraoral scanners. Studies have shown that the intraoral scanners demonstrate high accuracy both with natural teeth and implants[J1] [29,30].
Point 16: state here the limits of your study and report that further studies are needed with a longer follow-up time, more patients etc
The following part was added according to your suggestions
The results of the present study should be examined in a critical light because of the small study population and the small evaluation period. Despite the limitation of this study it can be concluded that the monolithic zirconia restorations fabricated with CAD/CAM technology is a viable option for the restoration of single posterior teeth. This study revealed no fracture of single tooth monolithic crowns and the success rate was high. However, long-term randomized control studies with big sample of patients are required to adequately document possible benefits of monolithic zirconia restorations in comparison to other treatments.
Point 17: you should add at least two more references at the end of your discussion when you state the limits of indirect digitization
PMID: 29471825 - PMID: 30112398 add these ones
References are now included.